# Randomised Control Trial Investigating the Efficacy of Meloxicam and Sodium Salicylate Non-Steroidal Anti-Inflammatory Drugs for Calf Cautery Disbudding

**DOI:** 10.3390/ani13111768

**Published:** 2023-05-26

**Authors:** Stephanie Prior, Nicola Blackie, John Fishwick, Sophie A. Mahendran

**Affiliations:** Royal Veterinary College, Pathobiology and Population Sciences, Hawkshead Lane, Hatfield, Hertfordshire AL9 7TA, UK; nblackie@rvc.ac.uk (N.B.); jfishwick@rvc.ac.uk (J.F.); smahendran@rvc.ac.uk (S.A.M.)

**Keywords:** calves, disbudding, NSAID, lying behaviour

## Abstract

**Simple Summary:**

The disbudding of calves is undertaken as a routine management practice. It is known to be a painful procedure for calves, and it is a legal requirement that calves are disbudded under local anaesthesia. It is also recommended that calves receive pain-relieving drugs (NSAIDs, non-steroidal anti-inflammatory drugs) which improve welfare and performance as well as reducing the pain experienced. Calves were allocated to one of two groups prior to undergoing disbudding; both groups received the same local anaesthetic. One group received a subcutaneous injection of Meloxicam as a positive control, and the second group received oral sodium salicylate added into their milk. Behaviour changes were monitored between treatment groups using accelerometers, with no differences being found in activity, lying bouts or lying times. There were also no differences in weight gain between the groups. Sodium salicylate offers a cheaper option for farmers when compared to meloxicam and can be administered orally, which is a less invasive technique than injection. Despite anaesthetic and NSAID administration, all calves showed behaviour changes for 5 days post procedure.

**Abstract:**

Disbudding calves using hot iron cautery is a routine management procedure to destroy the germinal cells around the horn bud in calves. It is recommended that NSAIDs are used in conjunction with local anaesthesia to reduce pain in calves during and after the procedure. In this study, two treatment groups were examined; calves in the positive control MEL group received subcutaneous meloxicam, and SAL calves received sodium salicylate orally for three days, both in addition to a local anaesthesia. Tri-axis accelerometers were attached to the calves, and DLWG (Daily Live Weight Gain) was measured. There was no significant difference between the treatment groups with regard to DLWG (*p* = 0.52), MI (motion index (*p* = 0.66)), lying bouts (*p* = 0.96) or lying times (*p* = 0.54). Given these findings, sodium salicylate may offer a lower-cost option for farmers when given at licensed doses compared to meloxicam, as well as providing a reduced-stress method of NSAID administration via an oral route. In addition, this study identified significant differences in activity in the time periods before and after disbudding, with MI (*p* < 0.01), lying bouts (*p* = 0.002) and lying times (*p* < 0.001) indicating changes in behaviour which extended to five days post disbudding.

## 1. Introduction

Within the UK, calf disbudding is a routine management practice conducted for the health and safety of both other animals and farm staff when handling cattle. The most common method utilised globally is hot iron cautery [1], which causes the thermal destruction of both the epidermal and dermal layers containing the germinal cells around the horn bud [2,3], resulting in third-degree burns on the disbudding site and second-degree burns on surrounding tissues [4]. In addition to the tissue damage around the horn bud, thermal cautery causes tissue damage and oedema that extends beyond the immediate horn bud, increasing the size of the sensitised area [5]. Following the initial procedure, the pain from disbudding is known to be present for an extended period afterwards, with the re-epithelialising tissue of a disbudding wound shown to be painful for 60 days or more post treatment [6], and with Carsoni et al. (2019) [7] suggesting that disbudding can induce chronic pain in animals due to damage to the cornual branch of the trigeminal nerve, leading to sensitisation and long-term pain.

In the UK, animal welfare legislation requires that local anaesthetic is used prior to hot iron disbudding [8]. This is normally applied in the form of a bilateral cornual nerve block approximately 10–15 min prior to disbudding [9]. Local anaesthetic is effective at providing a relatively short period (one to two hours) of pain mitigation [10,11] and delays the cortisol response indicative of pain-induced distress [12]. Other physiological parameters such as heart and respiratory rate have also been used as markers for pain in calves [13,14], with increases in both reported for calves that have undergone hot iron disbudding [2,15]. Eye temperature has also been monitored, with a rapid drop in eye temperature following disbudding due to sympathetically mediated vasoconstriction being virtually eliminated by the application of local anaesthetic [16]. Behavioural changes are also seen immediately following disbudding such as head shaking, ear flicking and head pressing, all indicating signs of pain in calves [2]. More recently, the use of accelerometer technology has enabled movement and play behaviour to be studied as a way of identifying pain or ill health in calves, with calves given pain relief at disbudding having longer lying times [13,14].

It is now accepted that further analgesia should be given when disbudding calves, with use of non-steroidal anti-inflammatory drugs (NSAIDs) reducing the longer-lasting inflammation-related pain associated with disbudding [2,17,18,19,20,21]. NSAID use significantly reduces behavioural changes following disbudding, with less head shaking, ear flicking and head rubbing [17], as well as increasing feed intakes [22] and growth rates [17,23]. This evidence has been used to encourage farmers to provide additional NSAIDs at disbudding, and in 2021, UK farm assurance guidelines were changed to include the requirement for calves undergoing disbudding to be given NSAIDs in addition to local anaesthesia (Red Tractor Certified Standards, London, UK). In order to provide additional analgesia, many UK farmers administer meloxicam subcutaneously to calves at disbudding [9], which has a plasma half-life of 26 h, helping to reduce physiological stress responses [18]. However, this does need to be injected into the calf, which in itself is a stressful and painful procedure. Sodium salicylate is an NSAID that has anti-inflammatory, analgesia and antipyretic effects [24,25], and is a soluble powder that can be administered in milk feeds [26]. This potentially allows for easy, less invasive, daily dosing of calves which may be more appealing to use for stockpeople. The objectives of this study were to establish if sodium salicylate administered orally was suitable for the alleviation of the pain associated with disbudding equivocal to that of meloxicam (used as a positive control) through the measurement of calf activity and growth rates. At the current time, sodium salicylate is not licenced for use for disbudding pain in the UK; the licenced use is for pyrexia in acute respiratory disease. An Animal Test Certificate was granted for its use in this study.

## 2. Materials and Methods

The study received ethical approval from the Royal Veterinary College (URN 2022 2132-3) and an ATC-S certificate from the Veterinary Medicine Directorate (ATC-S-197).

The study was conducted on a single-block calving dairy farm in North Devon between August and December 2022. Calves were housed in straw bedded igloos in groups of 14. A total of 59 calves were enrolled into the study, and all calves were born to a Jersey dam, with a mixture of jersey and beef breed sires. All calves were fed skimmed milk replacer, 22.5% protein and 25% Fat (Milkivit Energizer, Trouw Nutrition) fed at increasing quantities until the volume fed reached 6 litres per day split across two feeds offered in individual teat buckets. Weaning was conducted by reducing milk feeding to once per day at 10 weeks of age and stopped completely at 12 weeks of age. All calves had free access to water and a starter calf pellet (Harpers Calf Performer Nuts, 18% Crude Protein). Each pen of calves was systematically allocated to receive either a positive control, meloxicam, as their NSAID at disbudding (MEL), or sodium salicylate (SAL), such that the whole pen of calves received the same treatment. The meloxicam (Metacam, Boehringer Ingelheim Animal Health UK) was dosed at 0.5 mg/kg and injected subcutaneously at the same time as local anaesthetic administration prior to disbudding. The sodium salicylate (Solacyl, Dechra, Northwich, UK) was dosed at 40 mg/kg and was mixed into the milk feed of the calves. The first dose was given with the morning feeding of the calves (approximately 1 h prior to disbudding), with further dosing carried out by dosing at the morning milk replacer feed for 2 days after disbudding. The dosing regimen for both meloxicam and sodium salicylate were selected following the UK licenced dose rates and frequencies for each drug. The mean age of calves at disbudding was 35 ± 5 days. At disbudding, all calves received 5 mL of local anaesthetic, each side, in the form of Procaine (Pronestetic, Fatro, Bologna, Italy) administered as a cornual nerve block 10–15 min prior to disbudding. The efficacy of the nerve block was tested using a needle prick prior to application of the hot iron for cautery disbudding, with the absence of a response showing efficacy. All calves were disbudded by the same operator who restrained calves using head-locking yokes. Once disbudded, all calves were released from the head-locking yokes and were left to return to normal farm management.

Within each pen, 10 calves were fitted with tri-axis accelerometers (IceQube, Ice Robotics, Stirling, UK) applied to a hind limb at least 6 days prior to disbudding and left on for at least 6 days following disbudding. This allowed for the analysis of 11 days of activity data (5 days prior to disbudding, the day of disbudding, and 5 days following disbudding). This allowed the assessment of activity levels through lying times, lying bout number and motion index (activity) measurements. Calves also underwent weekly weighing between birth and 7 weeks of age via the application of a weigh tape (The Coburn Company, Whitewater, WI, USA) applied with calves in head-locking yokes during feeding time. The weigh tape measurements were performed by the same technician each week, who was blind to the previous weights. These measurements were used to calculate the daily liveweight gain DLWG (kg/day) for each week of life.

### Statistical Analysis

All data were stored in Excel (Microsoft Office; Microsoft, Redmond, WA, USA). All analyses were performed using SPSS (Version 27.0, IBM SPSS Statistics for Windows, NY: IBM Corp, Armonk, NY, USA). Significance was declared at *p* ≤ 0.05, and trends were reported if *p* ≤ 0.10. The outcome of the pre-weaning average daily liveweight gain (DLWG) was analysed using a generalised linear model with the categorical variables of treatment group (MEL or SAL), sex and breed. This was then further analysed to compare the DLWG in the week prior to, of and following disbudding using generalised estimating equations. The motion index, lying times and number of lying bouts of calves wearing pedometers was assessed using a linear mixed effects model. The overall fixed effects included were treatment group (MEL or SAL), sex, breed and day of the study. The pen and calf identification number were included as random effects. For all analyses, the assumption of normality was assessed through the visual inspection of histogram distributions.

## 3. Results

A total of 59 calves were recruited into the study (Table 1), with three calves identified as being polled and therefore did not require disbudding and were thus excluded. The sires of the remaining 56 calves were recorded as Jersey *n* = 36, British Blue *n* = 11, Holstein *n* = 5 and Wagu *n* = 4. No calves died within the study period. The overall mean DLWG across all calves was 0.80 kg/day (range 0.43–1.25 kg/day), with no significant effect of treatment group (*p* = 0.52, OR = 2.4, CI 0.947–1.114) or sex (*p* = 0.11, OR = 0.86, CI 0.722–10.35). The breed of the calf did have a significant impact (*p* = 0.006), with Holstein X Jersey calves having a significantly higher mean DLWG of 1.1 kg/day compared to the other breeds (mean DLWG 0.81–0.95 kg/day) (Figure 1). The DLWG in the week prior to, of and following disbudding was calculated for calves where weight data were available (*n* = 46). This demonstrated the same associations as the overall pre-weaning period, with no significant effect of treatment group (*p* = 0.35, OR = 1.03, CI = 0.95–1.11) or sex (*p* = 0.99, OR = 0.86, CI = 0.72–1.04). The breed of the calf also had a significant impact, with Holstein X Jersey calves having a higher mean DLWG of 1.3 kg/day compared to the other breeds, specifically in the weeks around disbudding (mean DLWG 0.84–0.96 kg/day, *p* < 0.001). Overall, the DLWGs continued to increase over the disbudding period, with the mean in the week prior to disbudding being 0.90 (SD 0.48) kg/day, the week of disbudding being 0.95 (SD 0.42) kg/day, and the week following disbudding being 1.04 (SD 0.42) kg/day.

Fifty-nine calves had accelerometers attached to their hind legs; however; due being polled, loss of accelerometers or technical errors, there were fifty calves with a full data set available for analysis. The motion index (MI) of the calves for the five days prior to disbudding and the five days following disbudding was not associated with the treatment group (*p* = 0.66, MEL group MI mean 6962 (CI = 6141–7784), SAL group MI mean 6791 (CI = 5766–7696) or breed (*p* = 0.46). However, the MI was significantly associated with the sex of the calf (F_1,45_ = 7.44, *p* = 0.009), with males being less active than female calves (MI for males of 5591, CI = 3968–7114, compared to females of 8112, CI = 7528–8777), and with the day of the study (F_10,473_ = 11.9, *p* < 0.01) which identified a reduction in MI on the days following disbudding (Figure 2).

The number of lying bouts across the 11-day observation period was not associated with the treatment group (*p* = 0.96), the breed (*p* = 0.86) or the sex of the calf (*p* = 0.89). It was significantly associated with the day of the study (F_10,474_ = 2.89, *p* = 0.002), which identified a reduction in lying bouts on the days following disbudding (Figure 3).

The lying times across the 11-day observation period were not associated with the treatment group (*p* = 0.54) or the sex of the calf (*p* = 0.16). It was significantly associated with the day of the study (F_10,474_ = 5.0, *p* < 0.001), which identified an increase in mean lying times on the days following disbudding. It was also significantly associated with the breed of the calf (F_1,12_._7_ = 6.27, *p* = 0.027), with dairy breeds having higher mean lying times than beef breeds (mean 17.4 h compared to 16.5 h) (Figure 4).

## 4. Discussion

This study compared the effects of two different non-steroidal anti-inflammatory drugs (NSAIDs) for use as pain relief for cautery disbudding. No negative control was used in this study due to the milk contract requirement for NSAID provision on the study farm, which is also widely accepted as best practice on all UK farms. Meloxicam was used as a positive control due to its common use across the UK, enabling comparison with sodium salicylate.

There appears to be limited studies that asses the impacts of NSAID combined with local anaesthesia on DLWG through the pre-weaning period, with previous work demonstrating improvements in short-term DLWG when NSAIDs were given without local anaesthesia [27,28]. Overall, this study found no significant difference between the NSAID treatment groups on the DLWG of calves throughout the study (*p* = 0.52). Given that previous studies have demonstrated the pain and stress mitigation effects of NSAIDs at disbudding [2,17], this suggests that any impact on feeding habits and weight gain were mitigated equally by both treatment protocol NSAIDs. Further analysis showed that DLWG continued to increase across both treatment groups in the week of disbudding (0.95 (SD 0.42) kg/day) and the week following disbudding (1.04 (SD 0.42) kg/day), again suggesting no difference in feed intakes between sodium-salicylate- and meloxicam-treated calves. This also suggests that there was no alteration in milk feeding due to the presence of the sodium salicylate within the milk replacer. To further clarify the impact of NSAIDs on DLWG, a negative control group would be needed to allow comparison with calves only receiving local anaesthesia and no form of NSAID treatment. It should be noted that the small sample size in this study (28 calves per group) meant that a post hoc power calculation using the observed weight difference found of 0.05 kg per day from this study had low power (20%), which may mean a biologically meaningful effect on the DLWG may have occurred. To establish whether the detected DLWG difference found in this study of 0.05 kg/day is significant at 80% power, 190 calves per group would be required.

This study used a weigh tape applied with calves in head-locking yokes during feeding time in order to minimise any handling stress, with tapes reported to offer a reliable proxy for calf weight [29]. Weighing the calves on calibrated scales would have provided more accurate weights; however, this would have involved further handling stress to the calves and may have impacted the results from the accelerometers. The same operator took all weights throughout the study to try and minimise inter-operator error, and the operator did not have previous weights available during weighing to reduce bias.

Tri-axis accelerometers were used to assess lying behaviour and activity levels in the calves. The motion index (MI) quantified how active the calves were [30], with the data indicating a significant reduction in MI in the days following disbudding (*p* < 0.01), but with no difference between the SAL and MEL treatment groups (*p* = 0.66). The reduction in MI after disbudding was likely due to the pain induced by the procedure as calf behaviour has been shown to alter after disbudding [2]. Similarly, lying time (*p* = 0.54) and number of lying bouts (*p* = 0.96) demonstrated no significant difference between treatment groups. This could indicate that there were no impacts of either NSAID on calf behaviour, or that both meloxicam and sodium salicylate affected behaviour to the same degree. A negative control where no NSAID was administered to a group of calves and directly compared to sodium salicylate would be needed to show any direct effects of sodium salicylate, but not administering any NSAIDs to calves at disbudding may raise ethical concerns. However, there was a significant reduction in both lying time and lying bouts in the five days following disbudding, with neither of these metrics returning to their pre-disbudding level during the study period. This indicates that the calves were negatively impacted for more than five days following disbudding despite the administration of local anaesthetic and additional analgesia. These findings are supported by other work which describes disbudded tissue as being more sensitive compared to non-disbudded tissue for at least three weeks after the procedure [6]. This may impact the calves’ willingness to initiate and engage in play behaviours following disbudding, leading to the reduction in MI and an increase in lying times. Directly observing the calves before disbudding and in the hours or days following may have demonstrated specific behaviours that reduce or altered in this study. There may also have been behaviours that altered in pattern, frequency or that were missing between the two treatment groups which may not have resulted in a change in the MI. Further research should be carried out to help to establish if the change in MI can be shown to be the equivalent of a change in behaviour.

Additional findings of interest were that male calves were less active compared to female calves (*p* = 0.009), which is consistent with other studies [31,32] and is likely due to innate activity differences between sexes. We also demonstrated that dairy cross calves had significantly higher mean lying times than beef cross calves (17.4 h compared to 16.5 h; *p* = 0.027) after disbudding. This could have been due to the smaller calf size of dairy cross breeds at disbudding, meaning that the handling and disbudding stress might have been higher in these animals. Further work would be needed to assess any differences in breed stress or activity at disbudding.

Overall, these findings raise important questions as to the longer-term pain management in calves undergoing disbudding procedures. It is currently unknown how long these behaviour changes take to resolve after disbudding, with a 5-day follow up in this study not resulting in a return to pre-disbudding activity levels. Most disbudding analgesia studies are focussed on the immediate period following disbudding; however, it is apparent that the effects on the calves are much longer lasting and not mitigated by current treatment protocols. If the normal moving and lying behaviours of the calves do not return to the pre-disbudding level for greater than five days following the procedure using standard treatment protocols, should additional anaesthesia at the time of disbudding or the longer-term provision of analgesia be offered? Meloxicam has been shown to be detectable in calves 50 h post treatment, although the effectiveness of the NSAID action over this duration was not demonstrated [28]. Sodium salicylate has a half-life of 1.17 +/− 0.26 h [33] and therefore requires more frequent dosing compared to meloxicam. In addition to the relative ease of oral dosing in milk feeds without the need to re-handle the animals, this could provide a potentially stress-free method of increasing NSAID treatment days or providing treatment well in advance of the disbudding procedure. Further investigations on the time taken for ‘normal’ or pre-disbudding activity levels are required to fully understand the impact that this routine husbandry procedure has on these animals.

A further area for consideration when treating animals with sodium salicylate may be altering frequency, duration or dose rates. Due to the relatively short half-life of sodium salicylate, twice-daily dosing in milk feeds may provide a greater consistency of analgesia over a 24 h period. Sodium salicylate can be administered in drinking water as a pulse medication and then can be administered twice daily under the UK licence. This could offer an alternative method of non-invasive NSAID treatment if it were shown to be effective. The potential for calves to not drink as expected following disbudding and the potential for unreliable dosing with some calves receiving a higher dose then others depending on thirst will need to be considered if dosing calves through water rather than milk. Free water intake can vary between calves, depending on age of calf and volume of cow’s milk or milk replacer fed [34], which may provide additional difficulties for dosing. As this study has demonstrated, behaviour is altered in calves for a minimum of 5 days post disbudding, and sodium salicylate could be continued for a longer period to offer analgesia for longer than the three days used in this trial. Increasing the duration of treatment may provide further analgesia, and alterations in behaviour may be seen at this treatment length; sodium salicylate has been used up to five days in a calf pneumonia trial without any reported ill effects [35]. The dose rate of 40 mg/kg was used in this trial as this is the dose rate licenced for calves in the UK, although the licence does not cover analgesia in calves following disbudding, and it is possible that a higher dose rate may be required against disbudding pain. An increase in dose rate up to 80 mg/kg for 5 days has been suggested as tolerated by calves [26], although how this impacts the analgesia provided is unknown. Further studies are required to understand how the dose rate and frequency may be altered to maximise the analgesia administered to calves following disbudding.

Economically, the cost of sodium salicylate treatments is less when compared to meloxicam, providing a potential financial justification for its use. Providing analgesia to an 80 kg calf at disbudding would currently cost a UK farmer GBP 1.52 (Metacam, Boehringer Ingelheim Animal Health UK Ltd., Woking, UK) compared to a three-day course of sodium salicylate costing GBP 0.77 (Solacyl, Dechra Veterinary Products, Leawood, KS, USA) (costs calculated from https://www.farmacy.co.uk/ (accessed on 23 March 2023) The dose rates and treatment regimens costed above are within the current licence for each medication in the UK and appear to offer comparable levels of analgesia in this study. This reduction in cost may be useful in increasing the uptake of additional NSAID analgesia on UK farms as well as making longer courses of NSAID treatment economically justifiable if the product was to become licenced. It is important to note than any alterations to dose rate, frequency or duration may alter the cost comparison, and further treatments could result in greater expense.

Further work needs to be carried out to establish other treatments that may be used for continued reductions in pain, discomfort and behaviour changes that these animals undergo for a routine management procedure experience. The administration of Xylazine has been suggested and is now used regularly as a method to sedate calves to reduce handling stress around disbudding. The administration of xylazine without other anaesthesia or analgesia has not been recommended [2]. Reedman et al. (2021) [36] demonstrated that calves disbudded under sedation with xylazine in addition to local anaesthetic and parenterally administered NSAID had a higher rate of play behaviour 24 h post disbudding compared to their non-sedated counterparts. Other studies have been completed to investigate novel and xylazine-sparing drug protocols, including the administration of levomethadone and ketamine [37], although the impacts on pain relief and behaviour are unknown. Negative control trials, where possible, will help to establish the direct impact of NSAIDs as opposed to comparing treatments.

Finally, the use of polled genetics is often discussed as a method of removing the need for disbudding altogether, although the adverse impact of polled genetics on the genetic merit of the calves is often raised as the main objection [38]. The use of gene editing may provide a way for the increased inclusion of polled genetics into the main reproductive pool, should legislation allow for this in the future. This would remove the need for disbudding calves entirely and improve calf welfare.

## 5. Conclusions

The use of oral sodium salicylate fed over three days showed no difference in DLWG, MI, lying time or number of lying bouts when compared to parenterally administered meloxicam. The results from this study show that the use of sodium salicylate has the potential to offer a cheaper and less stressful way to administer NSAIDs to calves undergoing disbudding. However, significant alterations in calf behaviour were noted in both treatment groups in the five days following disbudding, with calves demonstrating increases in lying times and reduced motion index, indicating that disbudding takes more than 5 days to recover from despite current protocols to provide analgesia.

## Figures and Tables

**Figure 1 animals-13-01768-f001:**
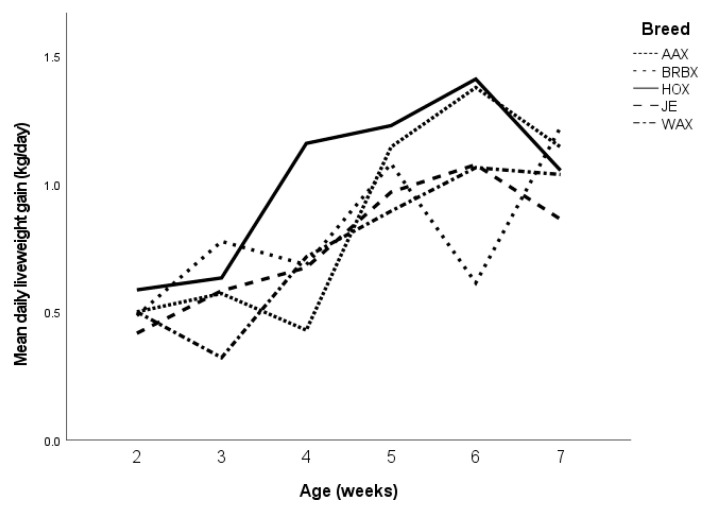
Comparison of the pre-weaning DLWG of the calves, shown by breed. Je = Jersey, HOX = Hotstein cross, AAX = Aberdeen Angus Cross, BRBX = British Blue Cross, WAX = Wagu Cross.

**Figure 2 animals-13-01768-f002:**
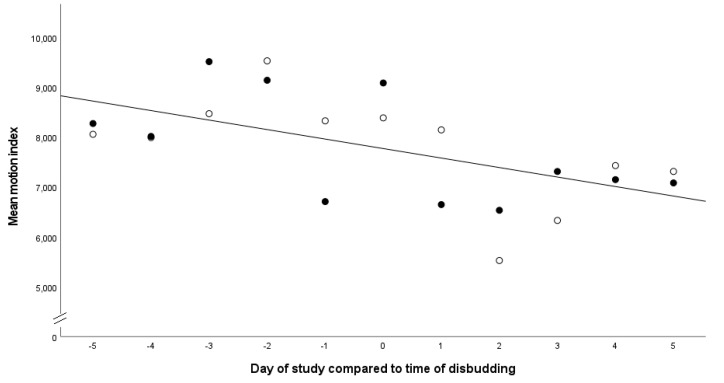
Comparison of the motion index of calves in the five days before and five days after disbudding. Solid black circles indicate calves in the SAL group, and white circles indicate calves in the MEL group. The line indicates the line of best fit. R^2^ 0.345 (y = 7770 − 1940 × x).

**Figure 3 animals-13-01768-f003:**
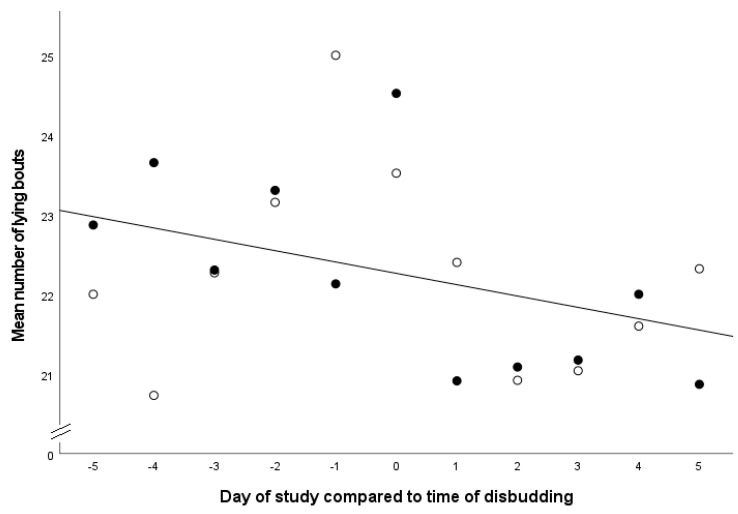
Comparison of the number of lying bouts of calves in the five days before and five days after disbudding. The solid black circles indicate calves in the SAL group, and the white circles indicate calves in the SAL group. The black line is the line of best fit. R^2^ 0.144 (y = 22.26 − 0.14 × x).

**Figure 4 animals-13-01768-f004:**
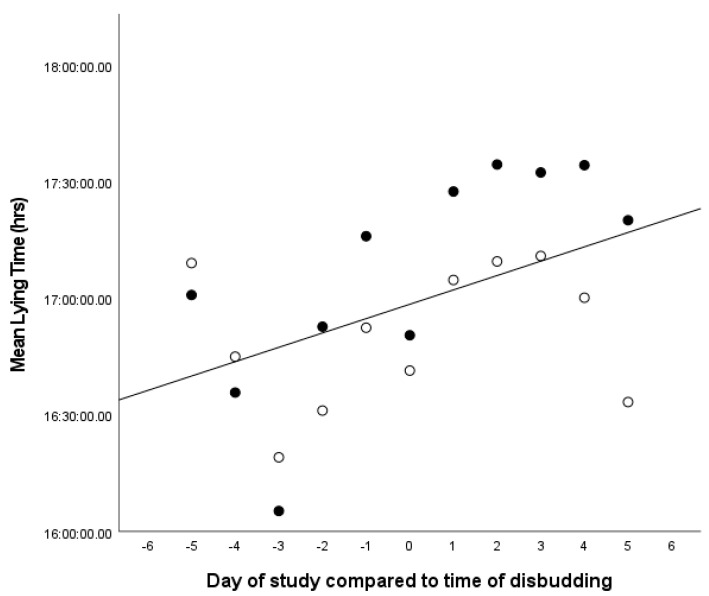
Comparison of the lying time (in hours) of calves in the five days before and five days after disbudding. Solid black dots indicate dairy breed calves, and white dots indicate beef cross breed calves. R^2^ 0.237, (y = 61,100 + 222 × x).

**Table 1 animals-13-01768-t001:** Overview of the pen signalments within the study. Dairy included Jersey and Jersey X Holstein. Beef included Jersey crossed with either Aberdeen Angus Cross, British Blue Cross or Wagu.

Pen	Sex (F:M)	Breed (Dairy:Beef)	Treatment Group (MEL or SAL)	Mean DLWG (Range), kg/day
1	10:0	10:0	SAL	0.88 (0.57–1.15)
2	10:0	10:0	MEL	0.94 (0.82–1.25)
3	10:0	10:0	SAL	0.68 (0.52–0.89)
4	10:0	10:0	MEL	0.74 (0.50–1.06)
5	9:1	3:7	SAL	0.76 (0.92–0.43)
6	4:5	0:9	MEL	0.79 (0.64–1.03)

## Data Availability

The data presented in this study are available on request from the corresponding author.

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
