# Peer review of "Randomised Control Trial Investigating the Efficacy of Meloxicam and Sodium Salicylate Non-Steroidal Anti-Inflammatory Drugs for Calf Cautery Disbudding"

_animals, 2023, doi:10.3390/ani13111768_

Round 1
Reviewer 1 Report
This paper compares the effects of meloxicam vs sodium salicylate for control of disbudding pain. I have a couple of suggestions.
Line 75-Are you sure this would be the right reference to say that meloxicam is preferentially a cox-2? Or is this a reference that cited the work of others?
Line 77-Do you have PK/PD evidence that sodium salicylate is an analgesic and antipyretic?
Line 94-You're missing a word in this sentence.
Line 102-106-How do you justify your treatment protocols? You only gave a single dose of meloxicam? Do you know the PD of the drug that justifies 3 days of analgesia? Additinally, later in the paper you talk about sodium salicylate having a very short half life. Based on that estimate, you probably should be dosing these calves BID to keep any levels of the drug in blood past 12 hours post-treatment. This should be better defended.
Lines 163-165-Be consistent with your P-value designations.
Line 221-I think the p-value quoted here is different than in the results. Or is this new data not presented.
Reviewer 2 Report
Comments attached as file

Round 2
Reviewer 1 Report
Thank you for the edits. I think you are missing a word in line 165.
Author Response
I think you are missing a word in line 165.’
This line now reads: ‘Fifty-nine calves had accelerometers attached to their hind legs…’
Reviewer 2 Report
I'm happy with your authors response but the paper is still written ignoring the response. GIven your study design, you need to be able to point to specific studies that demonstrate that meloxicam has an effect on the outcomes that you have measured - you haven't done that. Without those studies you cannot make any claim that salicylate is an 'effective alternative'. It might be but your study can only rule it out not rule it in.
We then add in the issue of power - what difference can your 'non-significant' study detect, can you rule out biogically important differences? There is no discussion of this.
So although the study has bee carried out well and has produced interesting results, I do not think it supports your interpretation. You need to be more circumspect - we don't know if the positive control was positive so the best you can say (if power was sufficient) is that for these outcomes there was no apparent diffeence which justifies further investigation of salicylate as a potential NSAID for use after (or perhaps even before) disbudding
Specific comments
25: No need for capitals with either meloxicam or sodium salicylate - remove throughout
29: This is very much 'may". These findings may not rule out a biologically important difference (due to the lack of power) and you don't know whether your positive control was a true positive control for these measures. For DLWG we know that meloxicam on its own will increase DLWG but not if given with sedation and LA (Bates et al 2016), so SS being the same as meloxicam with LA and sedation would be meaningless as nothing is as good (and cheaper and less stressful). All you can claim is that SS may be as good not that it is as good
79: your assumption is that daily dosing is appropriate, but as you stated for meloxicam analgesia is not the same as anti-inflammatory, so simply transferring pyrexia treatment to analgesia is not appropriate. It would also be useful to reference the daily dosing of salicylate for pyrexia - did the studies show it was optimal or just that it was 'effective'
107: Following
237: should be appear to not - but not sure how you can claim that without a negative control of no/sham disbudding
238: Again there may be no benefit of NSAIDs on DLWG in calves given local as the pain may still be there but not be sufficient to impact DLWG. You could then add in your lack of power which may mean that you can't be sure that a biologically meaningful effect could have occurred and that you missed it and you can't make these claims. The best that this study can do is show that SS may be as effective as meloxicam in mitigating disbudding pain but you need more negative control studies to do this
242: No - you had no negative control group
245: Post-hoc power analyses are very difficult to get right which is why many people don't like them. But post-hoc power is what many people do when planning a new study - take the results of the previous study and decide how many animals they need to identify a meaningful difference. That is what you need to do - what would be a meaningful difference? I would suggest a 10% difference in growth rate ~0.1 kg/day - how many calves (based on your data) would you need to do to repeat this. You could also see how much difference you could have detected with 80% power
This is ignoring the behaviour results which may also be under-powered and may actually provide a better measure of effectiveness – but again you don’t cite papers to show that meloxicam in a LA disbudding study does alter the behaviour you measured
330: Only when you almost certainly underdose SS in a study where you don't know whether your positive control actually impacted what you measured and where your power is limited
None
